# *Bscl2* Deficiency Does Not Directly Impair the Innate Immune Response in a Murine Model of Generalized Lipodystrophy

**DOI:** 10.3390/jcm10030441

**Published:** 2021-01-23

**Authors:** Ahlima Roumane, George D. Mcilroy, Arda Balci, Weiping Han, Mirela Delibegović, Massimiliano Baldassarre, Philip Newsholme, Justin J. Rochford

**Affiliations:** 1The Rowett Institute, University of Aberdeen, Foresterhill, Aberdeen AB25 2ZD, UK; a.roumane.18@abdn.ac.uk (A.R.); g.mcilroy@abdn.ac.uk (G.D.M.); 2The Aberdeen Cardiovascular and Diabetes Centre, University of Aberdeen, Foresterhill, Aberdeen AB25 2ZD, UK; m.delibegovic@abdn.ac.uk; 3Curtin Health Innovation Research Institute, School of Pharmacy and Biomedical Sciences, Curtin University, Perth, WA 6102, Australia; Philip.Newsholme@curtin.edu.au; 4Institute of Medical Sciences, University of Aberdeen, Foresterhill, Aberdeen AB25 2ZD, UK; r04ab17@abdn.ac.uk (A.B.); massimiliano.baldassarre@abdn.ac.uk (M.B.); 5Laboratory of Metabolic Medicine, Institute of Molecular and Cell Biology, Agency for Science, Technology and Research (A*STAR), Singapore 138673, Singapore; wh10@cornell.edu; 6Center for Neuro-Metabolism and Regeneration Research, Guangzhou Regenerative Medicine and Health Guangdong Laboratory, Guangzhou 510700, China; 7School of Laboratory Medicine and Life Sciences, Wenzhou Medical University, Wenzhou 325035, China

**Keywords:** *BSCL2*, seipin, congenital generalized lipodystrophy, immunity, macrophages

## Abstract

Congenital Generalized Lipodystrophy type 2 (CGL2) is the most severe form of lipodystrophy and is caused by mutations in the *BSCL2* gene. Affected patients exhibit a near complete lack of adipose tissue and suffer severe metabolic disease. A recent study identified infection as a major cause of death in CGL2 patients, leading us to examine whether *Bscl2* loss could directly affect the innate immune response. We generated a novel mouse model selectively lacking *Bscl2* in the myeloid lineage (LysM-B2KO) and also examined the function of bone-marrow-derived macrophages (BMDM) isolated from global *Bscl2* knockout (SKO) mice. LysM-B2KO mice failed to develop lipodystrophy and metabolic disease, providing a model to study the direct role of *Bscl2* in myeloid lineage cells. Lipopolysaccharide-mediated stimulation of inflammatory cytokines was not impaired in LysM-B2KO mice or in BMDM isolated from either LysM-B2KO or SKO mice. Additionally, intracellular fate and clearance of bacteria in SKO BMDM challenged with *Staphylococcus aureus* was indistinguishable from that in BMDM isolated from littermate controls. Overall, our findings reveal that selective *Bscl2* deficiency in macrophages does not critically impact the innate immune response to infection. Instead, an increased susceptibility to infection in CGL2 patients is likely to result from severe metabolic disease.

## 1. Introduction

Congenital Generalized Lipodystrophy (CGL) is a rare genetic disorder, where patients display a near complete lack of adipose tissue, resulting in insulin resistance, hepatic steatosis and hypertriglyceridemia [1]. The most severe form of CGL, CGL type 2 (CGL2), is caused by mutations affecting the protein seipin (encoded by *BSCL2*) [2]. Seipin is known to be a critical regulator of adipogenesis, as its loss prevents adipocyte differentiation in vitro [3,4] and adipose tissue development in vivo [5,6,7,8]. More recently, it has been revealed that seipin plays an important role as a scaffold protein, capable of binding numerous proteins that play critical functions in lipid droplet organization and triglyceride synthesis [9,10,11,12].

Rio Grande do Norte in Brazil has one of the highest rates of CGL2 prevalence. A recent study of CGL2 patients from this region revealed that this condition reduced lifespan by more than thirty years [13]. The authors found that one third of patients died as a result of liver disease, consistent with the severe metabolic dysfunction in these patients. Curiously, however, one third of patients died of infectious diseases. This led us to examine the role of seipin in the innate immune response, which has not previously been investigated and could contribute to the high incidence of deaths caused by infection.

Pathogen phagocytosis by macrophages and the progression and maturation of pathogen-containing phagosomes, a crucial event for the destruction of the pathogen, occurs in parallel with the formation of lipid droplets. Seipin has been shown to play important, evolutionarily conserved roles in lipid droplet biogenesis and dynamics in multiple cell types and species from yeast to man [14]. Within immune cells, lipid droplets synthesize and store inflammatory mediators and are considered structural markers of inflammation [15]. Interaction of lipid droplets with pathogen-containing phagosomes has been increasingly reported in response to infections and may contribute to destruction or contribute to the survival of the microorganism within host cells [16]. Thus, via altering the lipid droplet function, seipin could play an important role in the capacity of macrophages to respond appropriately to infections. To test this hypothesis, we ablated *Bscl2* specifically within the myeloid cell lineage of mice and characterized the innate immune response in this model and in bone-marrow-derived macrophages (BMDM) from global *Bscl2* knockout (SKO) mice.

## 2. Experimental Section

### 2.1. Animal Studies

Myeloid-specific *Bscl2* knockout (LysM-B2KO) mice were generated by crossing *Bscl2*^(fl/fl)^ mice [8] with *Bscl2*^(fl/wt)^ mice expressing Cre recombinase controlled by the *Lyz2* promoter. *Bscl2* knockout (SKO) mice were generated using a previously described method [17]. Briefly, fertilized *Bscl2*^(fl/wt)^ one-cell embryos were incubated ex vivo with TAT-Cre recombinase (#SCR508, Sigma, Gillingham, UK), then reimplanted into surrogate female dams. The resulting pups were screened by PCR to detect the deletion, and experimental colonies of SKO mice were then generated. All animal procedures were approved by the University of Aberdeen Ethics Review Board and performed under project license P94B395E0, approved by the UK Home Office under the Animals Scientific Procedures Act 1986. Unless stated otherwise, mice had *ad libitum* access to water and a standard chow diet (CRM (P) 801722, Special Diets Services).

### 2.2. Metabolic Studies

Fat and lean mass were measured using the EchoMRITM-500 body composition analyzer (Zinsser Analytic GmbH, Eschborn, Germany). Mice were injected with 1 mg/Kg of lipopolysaccharides (LPS, Sigma) by intraperitoneal injection. For glucose tolerance tests, 5-h fasted mice received intraperitoneal injections of 2 mg/g D-glucose (Sigma). Blood glucose was monitored between 0 and 120 min by glucometer readings (AlphaTrak^®^ II, Zoetisus, Parsippany-Troy Hills, NJ, USA) from tail punctures. Body temperatures were measured using a lubricated rectal probe inserted in mice maintained at standard housing temperatures both prior to and 3 h after LPS injections. Serum glucose levels were determined using the Glucose Colorimetric Assay Kit (Cayman Chemical, Ann Arbor, MI, USA); The insulin, Tnfa and Il-10 analysis was performed at the Core Biochemical Assay Laboratory (Cambridge, UK). The quantitative insulin sensitivity check index (QUICKI) was calculated as previously described [18]. QUICKI = 1/[log(I0) + log(G0)], where I0 is fasting insulin (µU/mL) and G0 is fasting glucose (mg/dL). QUICKI is a dimensionless index without units.

### 2.3. Gene Expression

RNA was extracted from tissues or cells using the RNeasy mini kit (Qiagen, Hilden, Germany), treated with DNase I (Sigma), then reverse-transcribed with M-MLV reverse transcriptase (Promega, Foster, CA, USA). Quantitative PCR was performed on the CFX384 TouchTM Real-Time PCR Detection System (BioRad, Watford, Herts, UK). The gene expression was normalized using the geometric mean of three stable reference genes (*Nono*, *Ywhaz* and *Hprt*) or *18s*. Sequences and details of all qPCR primers and assay probe sets are given in the Appendix A.

### 2.4. Bone-Marrow-Derived Macrophages (BMDM)

BMDM were isolated as described previously [19]. Cells were cultured and matured for seven days in DMEM supplemented with 10% Foetal Bovine Serum (FBS, ThermoFisher Scientific, Perth, UK), 20% L929 conditioned media, 100 U/mL penicillin, 100 mg/mL Streptomycin (ThermoFisher Scientific, UK), 1 mM sodium Pyruvate (Gibco, Grand Island, NY, USA), 1x MEM Non-essential Amino Acid (Sigma) and 0.25 mM β-mercaptoethanol (Sigma) in untreated Petri dishes. For the LPS treatment, BMDM were seeded in six-well tissue-culture plates and challenged with 100 ng/mL LPS for 4 h.

### 2.5. Bacterial Infection

*Staphylococcus aureus* SH1000 mCherry was kindly provided by Professor Simon Foster, Krebs Institute, University of Sheffield, UK. Bacteria were added to macrophages in Hank’s Balanced Salt Solution (Gibco) at a multiplicity of infection (MOI) of 5. At one hour post-infection, the cells were washed with PBS (Sigma-Aldrich, St. Louis, MO, USA) and incubated for 30 min in BMDM growth media supplemented with 100 μg/mL gentamicin (ThermoFisher Scientific, UK). The infected BMDM were maintained in 5 μg/mL gentamicin. At indicated times, cells were lysed in 0.1% Triton X-100 (Sigma-Aldrich), and serial dilutions were plated onto BHI agar to determine colony-forming units (CFU).

### 2.6. Immunofluorescence

Infected BMDM plated on glass coverslips were fixed with 4% paraformaldehyde (PFA, Agar Scientific, Stansted, UK) and permeabilized in 0.2% Triton X-100 (Sigma-Aldrich), 0.2% bovine serum albumin (BSA, Sigma-Aldrich) and 50 mM NH4Cl (Sigma-Aldrich). Cells were incubated with anti-LAMP-1 antibody (#1D4B, Developmental Studies Hybridoma Bank) and then Alexa Fluor^®^ 488 secondary antibody (Invitrogen, Carlsbad, CA, USA). Images were acquired on a PerkinElmer Spinning Disk confocal microscope linked to a Hamamatsu CMOS ORCA Flash 4.0 camera. An image analysis was performed using Volocity software.

### 2.7. Data Analysis

All data are presented as mean ± SEM and were analyzed by an unpaired two-tailed Student’s t test or two-way analysis of variance with a Bonferroni post-hoc test, as appropriate, using GraphPad Prism. A *p*-value < 0.05 was considered statistically significant.

## 3. Results

In order to examine the role of seipin deficiency in the macrophage function, we crossed *Bscl2*^(fl/fl)^ mice [8] with *Bscl2*^(fl/wt)^ mice expressing Cre recombinase controlled by the *Lyz2* promoter (LysM-Cre) to generate myeloid-specific *Bscl2* knockout (LysM-B2KO) mice (Figure 1A). The analysis of multiple tissues and bone-marrow-derived macrophages (BMDM) revealed that *Bscl2* mRNA levels were readily detectable in BMDM from control mice and similar to the levels in gonadal white adipose tissue (gWAT, Figure 1B). *Bscl2* mRNA expression was significantly reduced in BMDM from LysM-B2KO mice but was unaltered in other tested tissues, including liver, spleen and brown adipose tissue (BAT). A significant increase in the *Bscl2* mRNA expression was however observed in gWAT (Figure 1B). The characterization of male and female myeloid-specific *Bscl2* knockout (LysM-B2KO) mice revealed no significant differences in body weight (Figure 1C,D), fat mass (Figure 1E,F) or lean mass (Appendix A) when compared to littermate controls (CTRL). The glucose tolerance tests indicated that neither male nor female LysM-B2KO mice were glucose intolerant at 24 weeks of age (Figure 1G,H). To assess whether *Bscl2* deficiency plays a role in the inflammatory response to Toll-like receptor 4 (TLR4) activation, LysM-B2KO mice were injected with a sub-lethal dose of lipopolysaccharide (LPS, 1 mg/Kg). This led to a significant reduction in the body temperature in male but not female mice, with no genotype effect (Figure 1I,J). This sexually dimorphic response in body temperature in male versus female mice has been previously observed by others in other mouse models in adulthood [20]. As expected from previous studies [21], the LPS treatment caused a significant decrease in the blood glucose levels in male and female control mice (Figure 1K,L). This response was not altered by myeloid *Bscl2* deficiency. Similarly, the glucose tolerance was not significantly changed in male or female LysM-B2KO mice when compared to the controls three hours following LPS injections (Appendix A). The serum triglyceride levels were unaltered by LPS injection in both males and females and were equivalent in the control and LysM-B2KO mice (Appendix A).

Next, we performed a serum analysis to determine any effects of seipin loss on the inflammatory response in LysM-B2KO mice. Serum insulin levels were not significantly altered in male or female LysM-B2KO mice when compared to control mice before or after LPS injection, and the quantitative insulin sensitivity check index (QUICKI) revealed that LysM-B2 mice were not insulin-resistant when compared to controls following the in vivo LPS challenge (Table 1). TLRs present on the surface of macrophages can sense LPS and trigger the synthesis of proinflammatory cytokines in order to eliminate the pathogen [22]. LysM-B2KO mice displayed decreased levels of serum TNF-α when compared to controls, although this was not significant (Table 1). IL-10 levels showed greater fluctuations among males and females following LPS injection, without being significant (Table 1). Taken together, our data indicate that LysM-B2KO mice responded to the in vivo LPS challenge in a similar manner as their littermate counterparts.

To investigate further, BMDM were isolated from the male and female control and LysM-B2KO mice. As expected, male and female LysM-B2KO mice exhibited a significantly reduced *Bscl2* expression, but no change in the expression of anti-inflammatory (*Il-10*) or proinflammatory cytokines (*Tnfa*, *Il-6*, *Il-1b*) was observed in the seipin-deficient cells (Figure 2A,B). BMDM from female control and LysM-B2KO mice were also examined in the absence or presence of stimulation with 100 ng/mL LPS for four hours. The induction of *Il-10*, *Tnfa*, *Il-6*, *Il-1β*, *iNos*, *Mcp1* was unchanged by the loss of seipin in LysM-B2KO BMDM, although the induction of *Il-1α* was modestly but significantly greater (Figure 2C–I).

BMDM isolated from LysM-B2KO mice show significantly reduced but still clearly detectable levels of *Bscl2* expression (Figure 2A,B). It is possible that this arose from an incomplete LysMCre-mediated recombination, and we were unable obtain western blots to accurately determine the resulting seipin protein levels. Therefore, to examine the effect of a complete *Bscl2* deletion in this cell type, we next examined the innate immune response in macrophages isolated from global *Bscl2* knockout (SKO) mice. These mice were generated by incubating fertilized *Bscl2*^(fl/wt)^ one-cell embryos with cell-permeable TAT-Cre recombinase in culture before reimplantation into surrogate dams (shown schematically in Figure 3A). These mice had the same gene deletion as the SKO mice we described previously [8] but were generated by this alternative method. Like other SKO mice, they had a similar body weight but a significantly reduced fat mass and increased lean mass compared to control mice (Appendix A) along with elevated glycaemia in the fed state and fasting hyperinsulinaemia with a QUICKI analysis indicative of insulin resistance (Appendix A). The *Bscl2* expression was decreased by more than 98% in SKO macrophages when compared to control samples (Figure 3B). However, similar to observations in LysM-B2KO BMDM, the basal and LPS-stimulated expression of anti- and proinflammatory cytokines was not significantly different between the control and SKO BMDM (Figure 3C–I).

Next, we examined the intracellular fate and the ability of SKO macrophages to phagocytose a common bacterial pathogen. Control and SKO BMDM were infected with *Staphylococcus aureus* SH1000 expressing mCherry. After 1.5 h post-infection, bacteria were internalized into macrophages and found in lysosome-associated membrane protein-1 (LAMP-1) positive vacuoles (Figure 3J). Sixty to seventy percent of intracellular *S. aureus* were found to colocalize with LAMP-1 vacuoles in both control and SKO macrophages (Figure 3K). To determine whether SKO macrophages could effectively kill *S. aureus* once internalized, bacterial survival was assessed using a gentamicin protection assay. We found that *S. aureus* clearance was similar in control and SKO BMDM, with less than 6% of viable bacteria remaining 24 h after infection (Figure 3L,M).

## 4. Discussion

*BSCL2* gene mutations cause severe lipodystrophy, in which liver disease and infection are the main causes of death [13]. Liver disease can be linked to the severe hepatic steatosis in these patients [1]. It is less clear why infections are linked to *BSCL2* deficiency, but a direct effect in macrophages is credible. Seipin plays an important, evolutionarily conserved role in the biogenesis of lipid droplets [23], organelles that have been shown to regulate macrophage function and infection resolution, as has lipid metabolism [16,24]. In addition, *BSCL2*/seipin deficiency has been shown to induce ER stress [23], a process that has also been implicated in altered innate immunity and myeloid cell dysfunction in type 2 diabetes and atherosclerosis.

The findings presented here for the first time directly examined the consequence of *Bscl2* deficiency in myeloid cells. We found no impairment of the innate immune response in LysM-B2KO mice challenged with endotoxin. Moreover, macrophages isolated from LysM-B2KO or fully seipin-deficient SKO mice displayed no alteration in LPS-induced anti- or proinflammatory cytokine responses. Additionally, SKO macrophages were capable of pathogen recognition, engulfment, phagolysosome maturation and clearance when infected with *Staphylococcus aureus*. Whilst we observed no dramatic changes in macrophage function in this study, we examined only LPS-induced immune responses. This is a rather artificial surrogate for bacterial infection in vivo, and it remains possible that a different result may be observed with more clinically relevant and accurate models of sepsis. Nonetheless, we believe that our findings provide substantial evidence that *Bscl2* deficiency within macrophages does not directly impair the innate immune response. Therefore, it is unlikely that *BSCL2* deficiency in the myeloid lineage *per se* significantly contributes to an increased risk of infection in CGL2 patients.

In light of our findings, it would appear that any increased risk of death from infection in CGL2 is likely to result as being secondary to adipose tissue deficiency and the severe metabolic disease observed in this condition. For example, it has been observed that patients with congenital leptin deficiency have impaired immunity and increased rates of death from infections [25]. Leptin is now known to be a key regulator of the innate and adaptive immune responses, with leptin deficiency or resistance leading to the dysregulation of inflammatory responses and increased susceptibility of infectious disease (Reviewed in [26]). Patients with CGL2 have a near complete loss of metabolic and mechanical adipose tissues. Consequently, this results in significant decreases in the circulating levels of the adipose-secreted hormone leptin. Additionally, both hyperglycaemia and dyslipidaemia have been implicated in the impairment of the normal innate immune response and in macrophage dysfunction in type 2 diabetes [27]. Of note, SKO mice are hyperglycaemic but do not display the hypertriglyceridemia observed in *BSCL2*-deficient patients [28]. However, this can be modeled more accurately in SKO mice crossed to a dyslipidemic ApoE-null background [29]. Thus, a comparison of the response to infection in SKO versus SKO/ApoE-null mice may permit a dissection of the effects driven by hyperglycaemia and hyperlipidemia. It is possible to speculate about other reasons that may place CGL2 patients at an increased risk of infection. Seipin is highly expressed in the central nervous system, and there could be centrally driven effects on the immune function that suppress the immune function of these patients. Overall, a better understanding of the mechanisms involved in this phenomenon may establish suitable conditions to examine therapeutic interventions that could decrease the susceptibility to infection and the mortality rate amongst patients with CGL2.

## Figures and Tables

**Figure 1 jcm-10-00441-f001:**
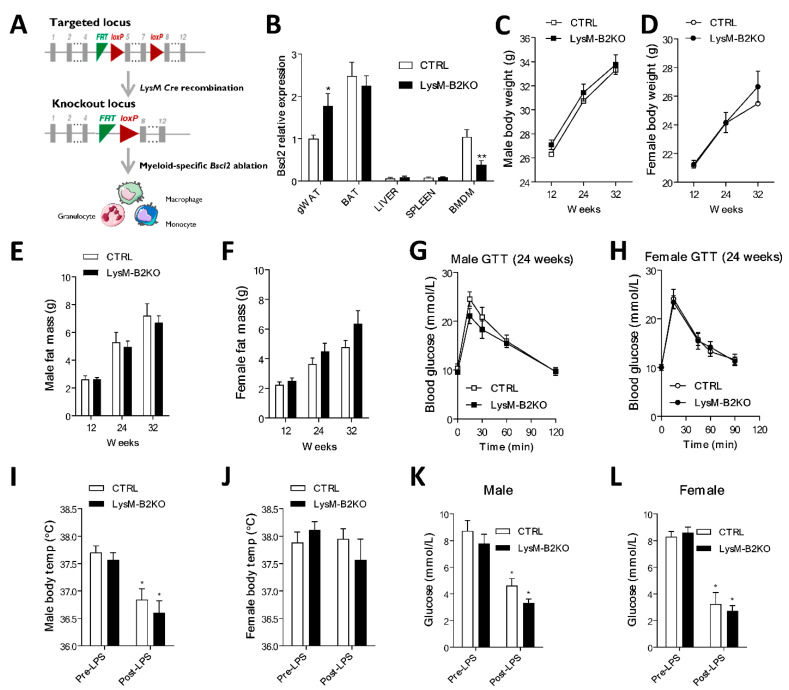
Physiological characteristics and lipopolysaccharide (LPS)-mediated inflammatory response in LysM-B2KO mice. (**A**) Targeting strategy for the conditional disruption of the *Bscl2* gene in the myeloid lineage. (**B**) *Bscl2* mRNA expression across different tissues and bone-marrow-derived macrophages (BMDM) of LysM-B2KO mice relative to the *18s* gene. (**C**,**D**) Male and female body weight, (**E**,**F**) male and female fat mass, and (**G**,**H**) male and female glucose tolerance of LysM-B2KO mice at 24 weeks of age. Effect of LPS on the (**I**,**J**) body temperature and (**K**,**L**) serum glucose in male and female LysM-B2KO mice at 32 weeks of age (female *n* = 5–7, male *n* = 7–10). Data are represented as mean ± SEM, * *p* < 0.05, ** *p* < 0.01.

**Figure 2 jcm-10-00441-f002:**
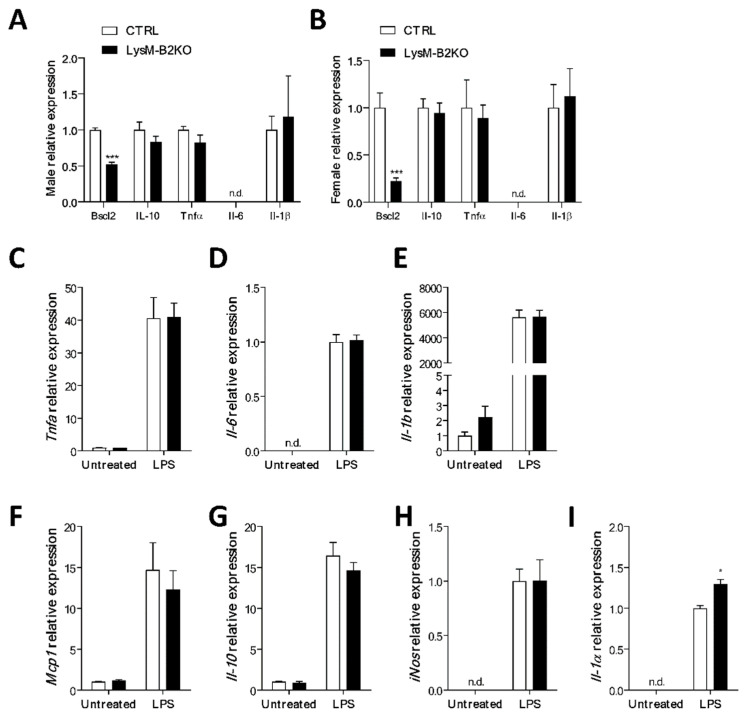
Lipopolysaccharides-mediated inflammatory responses in bone-marrow-derived macrophages. Basal mRNA levels of cytokines in BMDM isolated from (**A**) male and (**B**) female LysM-B2KO mice (female *n* = 6–7, male *n* = 6–9). (**C**–**I**) The treatment of female BMDM with LPS (*n* = 6) showed no significant difference in the expression of a panel of cytokines. Gene expression is normalized to three reference genes (*NoNo*, *Ywhaz* and *Hprt*). Data are represented as mean ± SEM, * *p* < 0.05, *** *p* < 0.001, n.d., not detectable.

**Figure 3 jcm-10-00441-f003:**
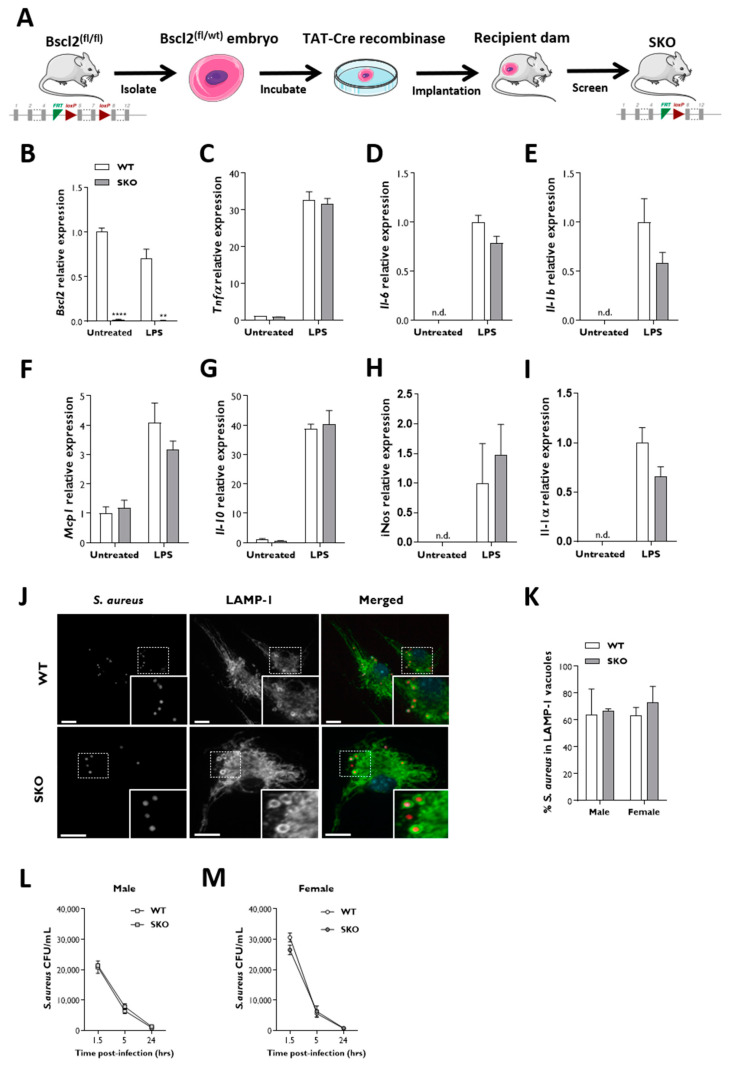
*Bscl2* knockout (SKO) mice BMDM efficiently respond to lipopolysaccharides and *Staphylococcus aureus* challenge. (**A**) Strategy for the generation of SKO global knockout mice. qPCR was performed to evaluate the expression levels of (**B**) *Bscl2* and (**C**–**I**) a panel of cytokines in female SKO BMDM challenged with LPS (*n* = 3). The gene expression was normalized to three reference genes (*NoNo*, *Ywhaz* and *Hprt*). Data are represented as mean ± SEM, ** *p* < 0.01, **** *p* < 0.0001, n.d., not detectable. (**J**) Representative confocal micrographs of the LAMP-1 distribution in BMDM infected with *S. aureus*-mCherry (*n* = 6). Dotted boxes indicate selected regions magnified in the white boxes. Bars denote 14 μm. (**K**) Quantification of the LAMP-1 association with *S. aureus*-mCherry. Data are represented as mean ± SEM. (**L**,**M**) Colony-forming units (CFU) of *S. aureus* inside male and female BMDM.

**Table 1 jcm-10-00441-t001:** Insulin, quantitative insulin sensitivity check index (QUICKI) and cytokine multiplex analysis of serum in LysM-B2KO mice fasted for 5 h and subjected to 1 mg/kg LPS for 3 h (female *n* = 5–7, male *n* = 7–10). Data are represented as mean ± SEM.

		Male		Female	
	Genotype	Pre-LPS Treatment	Post-LPS Treatment	Pre-LPS Treatment	Post-LPS Treatment
**Insulin (µg/L)**	**CTRL**	0.28 ± 0.15	1.75 ± 1.51	0.16 ± 0.03	0.21 ± 0.07
	**LysM-B2KO**	0.34 ± 0.13	0.99 ± 0.46	0.18 ± 0.11	0.39 ± 0.08
**QUICKI**	**CTRL**	0.34 ± 0.03	0.30 ± 0.04	0.36 ± 0.02	0.39 ± 0.04
	**LysM-B2KO**	0.33 ± 0.02	0.33 ± 0.02	0.37 ± 0.03	0.38 ± 0.02
**IL-10 (pg/mL)**	**CTRL**	-	453.91 ± 104.65	-	561.13 ± 113.92
	**LysM-B2KO**	-	404.87 ± 114.11	-	865.17 ± 408.16
**TNF-α (pg/mL)**	**CTRL**	-	390.78 ± 68.26	-	394.17 ± 387.63
	**LysM-B2KO**	-	313.94 ± 97.05	-	200.21 ± 107.88

## Data Availability

Data are available on request from the corresponding author.

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
