# Peer review of "Bscl2 Deficiency Does Not Directly Impair the Innate Immune Response in a Murine Model of Generalized Lipodystrophy"

_jcm, 2021, doi:10.3390/jcm10030441_

Round 1

Reviewer 1 Report

The manuscript by Roumane et al. tackles the question whether a myloid-specific loss of Bscl2, the gene responsible for congenital generalized lipodystrophy type 2 (CGL2), is responsible for a potential impaired immune response in patients with CGL2, since it has been shown that CGL2 patients frequently die from infection/pneumonia. The authors show that Bscl2 deficiency in myloid cells does not significantly impair innate immunity and cytokine response to LPS. Importantly, these mice had no apparent metabolic phenotype and normal fat mass. Unfortunately, the knock-down of Bscl2 in macrophages was incomplete. Therefore, the authors also looked at the immune response of bone marrow derived macrophages isolated from mice that globally lack Bscl2 and showed that these macrophages function similarly to those isolated from wild type mice. Thus, they concluded that the lack of Bscl2 in macrophages likely has no impact on the innate immune response.

The paper is well written, the experiments are methodologically sound and the data presented seems accurate and correctly interpreted. I also appreciate that both female and male mice were analyzed, which is not a given. Thus, I recommend this paper for publication in JCM. However, some issues have to be resolved before:

- The myloid-cell specific knock-out seems to be incomplete in the mRNA expression data presented in BMDMs. Can the authors pls show protein/seipin expression, since there could be different picture. If the protein expression data is similar, it is more accurate to use the term knock-down rather than knock-out.

- Have the authors also performed cecal ligation and puncture studies in these mice, since this is a more clinically relevant sepsis model compared to LPS injection. If not, pls comment and add to the discussion that LPS injection is a rather artificial model to assess an immune response to bacterial infection. This is a weakness of the manuscript that should be pointed out.

- add the primer sequences used for PCR to the method part and add the lot # of the antibodies used

Author Response

We are grateful for the positive comments regarding our manuscript and for the helpful suggestions regarding improvements we could make.

The myloid-cell specific knock-out seems to be incomplete in the mRNA expression data presented in BMDMs. Can the authors pls show protein/seipin expression, since there could be different picture. If the protein expression data is similar, it is more accurate to use the term knock-down rather than knock-out.

In common with other investigators, we have experienced several challenges western blotting for seipin protein. We have an excellent antibody from Cell Signalling for detecting the human protein but this does not detect murine seipin. We have previously used an antibody (AbCam #106793) to detect seipin in brown adipose tissue of wild-type versus SKO mice (Mcilroy et al., 2018, Mol. Met. 10:55-65). However, when examining several other tissues we find apparently non-specific bands of a similar size which are unaffected by Bscl2 ablation in our SKO mice. These appear to be absent in adipose tissues but they prevent us from detecting seipin reliably in some other tissues. This is particularly striking in liver samples. We see a similar effect in BMDMs, where none of the bands visible in control mice are absent in BMDMs from SKO mice, despite loss of the Bscl2 mRNA (as shown in Fig. 3A). We include a representative blot (please see the attachment) showing this phenomenon for the reviewer to illustrate the problem. Unfortunately, this means we are unable to determine the level of seipin protein reduction in myeloid-cell specific knockout mice. However, we have tried to clearly acknowledge that we cannot be sure how much the seipin levels are reduced in these cells. We and have further modified the text to clarify that our findings should be viewed with this caveat. It is for this reason that we performed the experiments with BMDMs from SKO mice.

Regarding the use of “knockout” rather than “knockdown”, we would be happy to do as directed by the editor. We appreciate the point the reviewer makes that there is clearly significant residual Bscl2 mRNA in the BMDM cell preparations, and have explicitly acknowledged this. Nonetheless the terminology “myeloid-specific Bscl2 knockout (LysM-B2KO)” is the accepted description of where a tissue specific Cre disruption of the gene. “Knockdown” would more typically indicate targeting of the mRNA, such as by siRNA. Hence, we feel that the current terminology is likely to be more clearly understood by the reader but would be happy to further edit the manuscript if requested to do so.

- Have the authors also performed cecal ligation and puncture studies in these mice, since this is a more clinically relevant sepsis model compared to LPS injection. If not, pls comment and add to the discussion that LPS injection is a rather artificial model to assess an immune response to bacterial infection. This is a weakness of the manuscript that should be pointed out.

The reviewer makes a very valid point and we fully accept that the LPS model is artificial, if commonly used. We hope they will appreciate that, as we did not observe a significant change in LPS response in Bscl2-deficient macrophages, we did not proceed further with more complex analyses for which we do not have the expertise. However, we recognise that this should be addressed and so have modified the manuscript to more explicitly acknowledge this as requested.

Add the primer sequences used for PCR to the method part and add the lot # of the antibodies used.

We have added the primers sequences in a supplementary table. Unfortunately, the antibody used to probe LAMP1 was not supplied with a lot number but the catalogue number is given in the Methods section.

Reviewer 2 Report

The study is well conducted, some minor revisions are needed.

Introduction, Line 55 "Within immune cells , lipid droplets synthesize and store inflammatory mediators and are considered structural markers of inflammation" , References are needed

Results line, 131: how the authors explain the significant reduction in body temperature in male mice but not in female one?

Discussion could be extended citing acquired lipodystrophies such as HIV-associated lipodystrophy that is linked to hepatic steatosis and dyslipidemia .

Authors should report other possible reasons for in risk of death from infection in patients with CGL2 other than leptin, hyperglyceamia and dyslipidemia. 

Author Response

We are grateful for the helpful comments of this reviewer and have added the reference requested and have revised our manuscript accordingly:

Results line, 131: how the authors explain the significant reduction in body temperature in male mice but not in female one?

Whilst we do not have a specific explanation of the underlying mechanism, this sexually dimorphic response has been observed by others in adult mice. We have noted this and added an appropriate reference in the text.

Discussion could be extended citing acquired lipodystrophies such as HIV-associated lipodystrophy that is linked to hepatic steatosis and dyslipidemia . Authors should report other possible reasons for in risk of death from infection in patients with CGL2 other than leptin, hyperglyceamia and dyslipidemia.

We have modified the discussion to address these points and hope that the reviewer finds this suitable. For the later point, we had tried to limit our speculations about alternative explanations of infection in CGL2. We have added a further comment regarding the possible involvement of the CNS. We would willingly add other suggested alternatives but were concerned about being overly speculative given we do not have data to support other possibilities.

Reviewer 3 Report

In the manuscript "Bscl2 deficiency does not directly impair the innate
immune response in a murine model of generalized lipodystrophy" Roumane et al., have shown that increased susceptibility to infection in congenital generalized lipodystrophy 2 patients is likely to result from severe metabolic disease. Using both whole-body Knock-out and myeloid cell lineage-specific Knockout of seipin the authors have shown that it is unlikely that BSCL2 deficiency in the myeloid lineage per se significantly contributes to increased risk of infection in CGL2 patient.  The authors have attributed the increased risk of death from infection in CGL2 is likely a cause secondary to adipose tissue deficiency and severe metabolic disease. The manuscript is written well and easy to understand and addresses an important question. The figure and figure legends are clear and easy to understand. The manuscript can be accepted as such with a minor spell check.

Author Response

We are grateful to the reviewer for their positive comments.